# Risk factors for excess all-cause mortality during the first wave of the COVID-19 pandemic in England: A retrospective cohort study of primary care data

Iain M. Carey◉*, Derek G. Cook◉, Tess Harris◉‡, Stephen DeWilde‡, Umar A. R. Chaudhry◉‡, David P. Strachan◉

Population Health Research Institute, St George's, University of London, London, United Kingdom

◉ These authors contributed equally to this work.
‡ TH, SD and UARC also contributed equally to this work.
* sgjd450@sgul.ac.uk

**Data Availability Statement:** The study is based on electronic health data from the Clinical Practice Research Datalink (CPRD) obtained under license

## Abstract

### Background

The COVID-19 pandemic's first wave in England during spring 2020 resulted in an approximate 50% increase in all-cause mortality. Previously, risk factors such as age and ethnicity, were identified by studying COVID-related deaths only, but these were under-recorded during this period.

### Objective

To use a large electronic primary care database to estimate the impact of risk factors (RFs) on excess mortality in England during the first wave, compared with the impact on total mortality during 2015–19.

### Methods

Medical history, ethnicity, area-based deprivation and vital status data were extracted for an average of 4.8 million patients aged 30–104 years, for each year between 18-March and 19-May over a 6-year period (2015–2020). We used Poisson regression to model total mortality adjusting for age and sex, with interactions between each RF and period (pandemic vs. 2015–19). Total mortality during the pandemic was partitioned into "usual" and "excess" components, assuming 2015–19 rates represented "usual" mortality. The association of each RF with the 2020 "excess" component was derived as the excess mortality ratio (EMR), and compared with the usual mortality ratio (UMR).

### Results

RFs where excess mortality was greatest and notably higher than usual were age >80, non-white ethnicity (e.g., black vs. white EMR = 2.50, 95%CI 1.97–3.18; compared to UMR = 0.92, 95%CI 0.85–1.00), BMI>40, dementia, learning disability, severe mental illness, place of residence (London, care-home, most deprived). By contrast, EMRs were comparable to

from the UK Medicines and Healthcare Products Regulatory Agency (MHRA). CPRD data governance and the license to use CPRD data does not allow distribution of patient data directly to other parties. Researchers must apply directly to CPRD for data access (https://www.cprd.com).

**Funding:** The author(s) received no specific funding for this work.

**Competing interests:** The authors have declared that no competing interests exist.

UMRs for sex. Although some co-morbidities such as cancer produced EMRs significantly below their UMRs, the EMRs were still >1. In contrast current smoking has an EMR below 1 (EMR = 0.80, 95%CI 0.65–0.98) compared to its UMR = 1.64.

## Conclusions

Studying risk factors for excess mortality during the pandemic highlighted differences from studying cause-specific mortality. Our approach illustrates a novel methodology for evaluating a pandemic's impact by individual risk factor without requiring cause-specific mortality data.

## Introduction

During the first six months of the global SARS-CoV-2 pandemic, England experienced its first wave of COVID-associated excess mortality, during which it was more severely affected than comparator countries in Europe and elsewhere [1]. During March to May 2020, there were almost 50,000 more deaths from any cause than would be expected at this time of year [2]. Concerns were expressed at the time [3] that part of the excess all-cause mortality may have been due to delays in presentation of non-COVID medical emergencies (such as heart attacks and strokes) during the pandemic lockdown period [4, 5].

Risk factors for death with COVID-19 as a certified cause up to early May 2020 have been analysed using electronic health records from a large set of English general practices assembled by the OpenSAFELY initiative [6]. In a later publication, extending the follow-up period to November 2020, the same group compared risk factors for COVID-19 deaths with those for non-COVID-19 mortality [7]. However, early in the pandemic, certification of deaths did not require confirmation of SARS-CoV-2 infection as virological testing was incomplete at this stage [8]. Thus, COVID-19 as a cause of death could also reflect a risk factor for admission or testing at this time. Only by placing results in a wider historical context, by comparing with pre-pandemic years, can a more comprehensive assessment of risk factors on excess and non-excess mortality be achieved.

In this paper, we use electronic primary health care records (different from those analysed by OpenSAFELY) to carry out a retrospective cohort study with an analysis that partitions the all-cause mortality during England's first 2020 wave into two components: "usual" (or non-excess) mortality, estimated from age-adjusted mortality in the same season in five pre-pandemic years; and "excess" mortality. We describe a novel statistical method for comparing socio-demographic, lifestyle and medical correlates of each component and compare our findings to the published results from OpenSAFELY [7, 9].

## Methods

### Data source

The Clinical Practice Research Datalink (CPRD) is a large primary care database in the UK jointly sponsored by the Medicines and Healthcare products Regulatory Agency and the National Institute for Health Research [10]. It provides a longitudinal medical record for all registered patients (where in the UK >99% of the population are registered with a GP), with diagnoses and other clinical information recorded on the system using Read codes [11]. CPRD now includes EMIS (Egton Medical Information Systems) practices (CPRD Aurum [12]),

resulting in a much larger dataset (>1,800 combined practices, 65 million patient lives, of which 16 million are currently active). The majority of contributing CPRD practices in England have consented to their data being linked to external sources, which is facilitated by a "trusted third party" to CPRD, ensuring that researchers have no access to geographical identifiers such as residential postcode [13]. Key variables which have been linked to the practice data include the Index of Multiple Deprivation (IMD), a composite small-area (approximately 1500 people) measure used in England for allocation of resources [14].

## Definition of annual cohorts

We first identified a set of practices within CPRD Aurum that were continually providing data to CPRD from 1st January 2015 to 1st August 2020 and had consented to data linkage. A total of 770 (56%) practices were identified, with exclusions due to data not available to August 2020 or no linkage available. From these practices, we then used patient registration dates to create annual cohorts of patients who were active in similar time periods in each of the 6 years (2015 to 2020). For this analysis of the first wave of the pandemic, the selected period in each year was 18th March to 19th May inclusive (corresponding to Weeks 11 to 20). Patients were only included once they had accrued 90 days of registration time, and their total registration time was counted in each year. We further restricted to adults aged between 30 and 104 years old, as there would be little excess mortality in the young as well as incomplete data for many risk factors, and also excluded a small number of patients (<1%) without linkage to IMD.

From each patient record, we extracted medical history, focusing on conditions routinely collected as part of the Quality and Outcomes Framework [15], which we had previously shown to be predictive of mortality [16]. The term "Mental Health" encompasses severe mental health disorders: psychosis, schizophrenia and bipolar affective disorder. Additionally, we extracted information on ethnicity, smoking, body mass index (BMI) and whether they were recorded as living in a care home. For each year the patient was eligible to be in analysis, concurrent variables were created based on the information recorded up to that point in time. Thus, a patient could be a smoker in one cohort but an ex-smoker in a subsequent one. The only exception was for ethnicity where a recording anywhere in the record was utilised to determine status (summarised as White, Black, South Asian, Mixed or Not Recorded).

While external linkage to national death certification data is available within CPRD, there is usually a time lag (up to one year) on its availability. To be able to study mortality into 2020, we therefore decided to only use mortality related information from the primary care record—either a relevant de-registration flag or a Read code. While there is near agreement between the CPRD and linked data, with over 98% of deaths in national mortality data reported to be also identified in CPRD [17], the CPRD date of death may be up to a month later than the actual date of death. However preliminary analyses comparing 2020 mortality rates with 2015–9 rates in our data suggested a weekly pattern of excess mortality similar to national figures for England (S1 Fig).

## Statistical methods

Stata version 15 (StataCorp. 2017. Stata Statistical Software: Release 15. College Station, TX: StataCorp LLC.) was used for statistical analysis. Each of the six 9-week periods (five pre-pandemic, one pandemic) was considered as a statistically independent "risk set" (since death can only occur once). All six periods were therefore combined into a single dataset for log-linear Poisson regression modelling, using death from any cause as the outcome, person-time at risk as an offset (S1 Appendix) and robust standard errors. All models were adjusted for age (as a linear term) and sex and included a dichotomous term for the pandemic (2020) v pre-

pandemic years (2015–2019 combined). Other risk factors were individually added to this basic model, with interaction terms included to assess effect modification, contrasting pandemic and non-pandemic periods. Thus, age-sex-adjusted total mortality rates were modelled as a combination of usual exposure-specific effects (U), pandemic-associated effects (P) and the statistical interaction between the two (I).

The modelled risk factor effects on total mortality were partitioned into patterns of association with usual mortality (estimated directly from the pre-pandemic years and termed the usual mortality ratio (UMR)) and with excess mortality. The ratio of the excess mortality rates between exposed and non-exposed provides an estimate of the true pandemic interaction (T) for excess mortality (see box/appendix for details). By multiplying T * U, this then provides an estimate of the relative effect of exposure on excess deaths, which we have termed the excess mortality ratio (EMR).

We used the *lincom* command in Stata to derive 95% confidence intervals for T and EMR. The modelled interaction term (I) is scaled by the appropriate factor (In[T]/In[I]) in the *lincom* statement that produces estimates for T and EMR, as well a 95% confidence interval for each which assumes that the derived Wald Test (and z-score) from the scaled model interaction is the same as that estimated using the (unscaled) model interaction (I). Thus, we are assuming that the error largely arises from the error in estimating the original interaction term, which seems reasonable since it represents a reparameterization of the same model.

Our main models only adjusted for age and sex as we were primarily interested in comparing effects between periods (pandemic and non-pandemic) and the identical sampling structure in the data would produce similar cohorts by periods. However, sensitivity analyses also fitted models that included further overall adjustment (smoking, BMI, ethnicity, IMD, region) to investigate the impact on the estimates of T and EMR. Due to the very strong effects of age on mortality in both pre-pandemic and pandemic periods, and evidence of a markedly steeper age-related gradient during the pandemic, further sensitivity analyses fitted age-stratified models for each risk factor, using three age groups for presentation (30–64, 65–79, 80 or more years). Additionally, for some co-morbidities, we subdivided into care-home residents and other persons, to address specific concerns about the prominence of this wave of mortality among care homes throughout the UK.

## Ethics approval

The protocol (no. 20_148) was approved by the Independent Scientific Advisory Committee evaluation of joint protocols of research involving CPRD data in July 2020. The approval allows analysis of anonymous electronic patient data without the need for written or oral consent.

## Results

### Study population

The study population identified (adults aged 30–104 years) grew from 4.4 million in 2015 to 4.8 million to 2020 (Table 1). The number of deaths identified on the database was between 10,000–10,500 in usual years, and 16,735 in 2020. Adjusting for age and sex, we estimated that CPRD patients were 51% (95%CI 49–54%) more likely to have died during 2020 compared to identical periods (18th March to 19th May) than in the last 5 years. This equates to approximately 5,800 excess deaths in our dataset. A summary of the risk factor recording is provided in S1 Table. In 2020, >80% of patients had an ethnicity recorded, >90% a BMI and 98% smoking status.

**Table 1. Summary of annual patient cohorts based on 18th March to 19th May dates.**

| Years | Total patients | Total Deaths | % |
|---|---|---|---|
| 2015 | 4,411,547 | 10,514 | 0.24% |
| 2016 | 4,487,997 | 10,407 | 0.23% |
| 2017 | 4,577,321 | 10,070 | 0.22% |
| 2018 | 4,661,805 | 10,458 | 0.22% |
| 2019 | 4,757,897 | 10,183 | 0.21% |
| 2020 | 4,835,708 | 16,735 | 0.35% |

The geographical regions of the 770 CPRD Aurum practices used in the data were: North West = 143, North East = 41, Yorkshire & Humber = 23, West Midlands = 171, East Midlands = 19, East of England = 30, South West = 84, South Central = 69, South East Coast = 59, London = 131.

## Overall findings—Age and sex

Table 2 and Fig 1 summarise the EMR estimates by risk factors in the overall study population derived from a model that adjusts for age and sex. The overall mortality ratios for 2020 are presented, with the subsequent partition into the UMR and the EMR. The formal comparison of the EMR with the UMR ("True Pandemic Interaction") is given in the final column of the table, with a corresponding symbol on the figure denoting whether the excess or usual mortality ratio was significantly higher. Note that age only appears in the table as the plotted mortality ratios would require a change of scale to interpret visually on the plot.

While men were 38% more likely to have died during 2020 than women (after adjusting for age), this was only marginally higher than the usual observed estimate (34%). Thus, the estimated EMR of 1.46 was not significantly higher than the UMR (true pandemic interaction = 1.09, 95%CI 0.98–1.20). Age was summarised into 10-year age groups with 70–79 years as the reference category to facilitate an easier interpretation. This reveals that for age groups <70 years, mortality in 2020 was lower than would be expected. However, for ages ≥80 years, the opposite was true, and significantly higher EMRs were estimated.

## Risk factors with significantly greater excess mortality

Risk factors where excess mortality was greatest and notably higher than usual were: all non-white ethnicities, BMI>40 and place of residence (London, most deprived, care home). For example, people of black ethnicity versus white had an EMR = 2.50 (95%CI 1.97–3.18), while for Asian ethnicity the EMR = 1.50 (95%CI 1.19–1.90) compared to white. As non-white ethnicities had UMRs<1 in pre-pandemic years, the estimated true pandemic interactions were higher still (Black = 2.72, Asian = 1.87). For BMI, both low values (<20) and very high values (>40) produced EMRs in excess of 2.7. However, comparing these to the UMR, an additional impact of the pandemic is only observed for the morbidly obese (>40) group.

While 2020 mortality showed a clear trend with IMD, this was not too dis-similar to the usual trend, and as a result only the EMR for the most deprived quintile (EMR = 2.05, 95%CI 1.76–2.38) was significantly slightly higher than expected (UMR = 1.70, 95%CI 1.65–1.75). Among co-morbidities, the EMRs for dementia (9.87, 95%CI 9.00–10.82) and learning disability (8.54, 95% CI 5.99–12.18) stood out, though significantly higher estimates of EMR than UMR were also observed for chronic kidney disease, diabetes, epilepsy, hypertension, severe mental illness, osteoarthritis and stroke.

**Table 2. Mortality ratios for 2020 and 2015–9 (Usual) with corresponding excess mortality ratio (EMR) and true pandemic interaction (TPI).**

| | 2020 Mortality Ratio (95% CI) | 2015–9 Usual Mortality Ratio (UMR) (95%CI) | 2020 Excess Mortality Ratio (EMR) (95%CI) | True Pandemic Interaction[a] (95%CI) |
|---|---|---|---|---|
| **Sex** | | | | |
| • Females | 1 | 1 | 1 | 1 |
| • Males | 1.380 (1.339,1.423) | 1.342 (1.319,1.366) | 1.456 (1.324,1.602) | 1.085 (0.979,1.202) |
| **Age** | | | | |
| • 30 to 39 | 0.021 (0.018,0.025) | 0.025 (0.023,0.027) | 0.012 (0.006,0.025) | 0.486 (0.224,1.055) |
| • 40 to 49 | 0.056 (0.050,0.062) | 0.065 (0.061,0.068) | 0.035 (0.022,0.056) | 0.547 (0.336,0.890) |
| • 50 to 59 | 0.139 (0.129,0.149) | 0.146 (0.141,0.152) | 0.122 (0.093,0.160) | 0.833 (0.626,1.109) |
| • 60 to 69 | 0.368 (0.348,0.390) | 0.386 (0.375,0.399) | 0.328 (0.266,0.403) | 0.848 (0.679,1.058) |
| • 70 to 79 | 1 | 1 | 1 | 1 |
| • 80 to 89 | 3.532 (3.390,3.679) | 3.254 (3.178,3.331) | 4.151 (3.646,4.726) | 1.276 (1.109,1.467) |
| • 90 to 104 | 10.498 (10.029,10.988) | 9.327 (9.084,9.577) | 13.104 (3.321,9.665) | 1.405 (1.207,1.635) |
| **Smoking** | | | | |
| • Never | 1 | 1 | 1 | 1 |
| • Ex | 1.280 (1.238,1.324) | 1.274 (1.249,1.300) | 1.291 (1.172,1.422) | 1.013 (0.912,1.125) |
| • Current | 1.638 (1.556,1.724) | 2.123 (2.067,2.181) | 0.796 (0.647,0.979) | 0.375 (0.301,0.466) |
| **Ethnicity** | | | | |
| • White | 1 | 1 | 1 | 1 |
| • Black | 1.472 (1.326,1.635) | 0.920 (0.849,0.996) | 2.503 (1.969,3.181) | 2.721 (2.057,3.599) |
| • Asian | 1.049 (0.958,1.149) | 0.805 (0.755,0.858) | 1.504 (1.193,1.896) | 1.868 (1.437,2.422) |
| • Mixed | 1.307 (1.128,1.515) | 0.924 (0.829,1.030) | 2.021 (1.411,2.896) | 2.187 (1.447,3.306) |
| • Other | 1.162 (1.016,1.329) | 0.874 (0.793,0.962) | 1.700 (1.212,2.383) | 1.945 (1.324,2.860) |
| **BMI** | | | | |
| • <20 | 2.704 (2.578,2.836) | 2.689 (2.616,2.764) | 2.734 (2.354,3.176) | 1.017 (0.865,1.194) |
| • 20–30 | 1 | 1 | 1 | 1 |
| • 30–35 | 1.004 (0.956,1.053) | 0.959 (0.933,0.987) | 1.091 (0.942,1.203) | 1.137 (0.970,1.332) |
| • 35–40 | 1.254 (1.164,1.351) | 1.200 (1.147,1.255) | 1.361 (1.082,1.711) | 1.135 (0.885,1.455) |
| • 40+ | 2.200 (2.012,2.407) | 1.896 (1.791,2.007) | 2.801 (2.176,3.605) | 1.477 (1.119,1.951) |
| **Deprivation** | | | | |
| • IMD1 (Least) | 1 | 1 | 1 | 1 |
| • IMD2 | 1.175 (1.122,1.230) | 1.130 (1.101,1.160) | 1.271 (1.094,1.475) | 1.124 (0.958,1.321) |
| • IMD3 | 1.267 (1.209,1.328) | 1.223 (1.191,1.256) | 1.362 (1.171,1.585) | 1.114 (0.947,1.311) |
| • IMD4 | 1.433 (1.367,1.503) | 1.376 (1.339,1.414) | 1.556 (1.334,1.815) | 1.131 (0.958,1.334) |
| • IMD5 (Most) | 1.809 (1.724,1.898) | 1.698 (1.652,1.745) | 2.045 (1.757,2.380) | 1.205 (1.023,1.418) |
| **Region** | | | | |
| • London vs. Rest | 1.263 (1.209,1.320) | 0.974 (0.947,1.001) | 1.885 (1.672,2.122) | 1.936 (1.696,2.206) |
| **Care Home** | | | | |
| • Yes vs No | 4.154 (3.951,4.367) | 2.630 (2.539,2.725) | 7.547 (6.681,8.524) | 2.869 (2.500,3.292) |
| **Co-morbidities** | | | | |
| • Atrial Fibrillation | 1.669 (1.608,1.732) | 1.744 (1.706,1.782) | 1.524 (1.351,1.718) | 0.874 (0.768,0.994) |
| • Asthma | 1.120 (1.072,1.171) | 1.156 (1.127,1.186) | 1.052 (0.911,1.214) | 0.910 (0.780,1.062) |
| • Cancer | 2.031 (1.962,2.103) | 2.579 (2.528,2.630) | 1.121 (0.985,1.275) | 0.435 (0.379,0.498) |
| • Coronary Heart Dis. | 1.468 (1.415,1.524) | 1.444 (1.413,1.475) | 1.516 (1.353,1.698) | 1.050 (0.929,1.186) |
| • Chronic Kidney Dis. | 1.478 (1.430,1.528) | 1.332 (1.305,1.358) | 1.794 (1.626,1.981) | 1.348 (1.212,1.499) |
| • COPD | 2.005 (1.919,2.094) | 2.250 (2.196,2.306) | 1.547 (1.330,1.799) | 0.687 (0.585,0.808) |
| • Dementia | 4.817 (4.650,4.989) | 2.993 (2.924,3.064) | 9.870 (9.004,10.820) | 3.298 (2.980,3.649) |

*(Continued)*

**Table 2.** (Continued)

| | 2020 Mortality Ratio (95% CI) | 2015–9 Usual Mortality Ratio (UMR) (95%CI) | 2020 Excess Mortality Ratio (EMR) (95%CI) | True Pandemic Interaction[a] (95%CI) |
|---|---|---|---|---|
| • Diabetes | 1.697 (1.639,1.756) | 1.492 (1.461,1.523) | 2.144 (1.932,2.379) | 1.438 (1.284,1.610) |
| • Epilepsy | 2.264 (2.083,2.462) | 2.039 (1.939,2.143) | 2.708 (2.121,3.447) | 1.328 (1.021,1.722) |
| • Heart Failure | 2.349 (2.246,2.456) | 2.358 (2.296,2.422) | 2.329 (2.022,2.683) | 0.988 (0.848,1.150) |
| • Hypertension | 1.210 (1.173,1.249) | 1.053 (1.034,1.072) | 1.606 (1.452,1.775) | 1.525 (1.370,1.698) |
| • Learning Disability | 5.295 (4.580,6.121) | 3.645 (3.292,4.036) | 8.540 (5.986,12.183) | 2.343 (1.564,3.509) |
| • Mental Health | 3.068 (2.829,3.368) | 2.460 (2.326,2.601) | 4.276 (3.432,5.464) | 1.738 (1.361,2.218) |
| • Osteoarthritis | 1.476 (1.414,1.541) | 1.283 (1.249,1.319) | 1.878 (1.660,2.123) | 1.463 (1.219,1.673) |
| • Rheumatoid Arthritis | 1.448 (1.326,1.600) | 1.418 (1.341,1.499) | 1.506 (1.147,2.046) | 1.062 (0.789,1.479) |
| • Stroke or TIA | 1.842 (1.771,1.917) | 1.686 (1.646,1.726) | 2.158 (1.920,2.429) | 1.281 (1.122,1.454) |

[a] Defined as the ratio of the EMR to the UMR (see box). Note that all models adjust for age and sex.

## Risk factors with significantly lower excess mortality

Among the co-morbidities, cancer and Chronic Obstructive Pulmonary Disease (COPD) were notable in that they produced EMRs significantly lower than their UMRs. While both were still associated with an approximate doubling of mortality risk in 2020 (cancer = 2.03, COPD = 2.01), their estimated EMRs were between 1 and 2 (cancer = 1.12, COPD = 1.55) indicating that the true pandemic interaction ratio for patients with these conditions was <1 (cancer = 0.44, COPD = 0.69). Even more extreme was current smoking which was inversely associated with excess mortality: while current smokers were 64% still more likely to die than non-smokers in 2020, this was well below the UMR = 2.12 (95%CI 2.07–2.18), and hence the EMR was below one 0.80 (95%CI 0.65–0.98).

## Further adjusted and stratified analyses

Sensitivity analyses that included additional adjustment for other co-factors (S2 Table) generally attenuated the estimates for EMR, but significant effects of the pandemic were still observed for the same factors. For example, the EMR for black ethnicity fell to 2.20 (95%CI 1.74–2.80), with a true pandemic interaction of 2.61 (95%CI 1.97–3.46).

Stratified analyses by age group (30–64, 65–79, 80+) and care home (yes or not recorded) were also carried out. S3 Table stratifies the model estimates for sex, smoking, ethnicity, deprivation, BMI and region by age group (30–64, 65–79, 80+). People of Asian ethnicity under age 65, and of Black ethnicity under age 80 had a true pandemic interaction of >3.5, suggesting the impact of excess mortality within these ethnicities was more pronounced at younger ages. For BMI, among the under 65 olds, being morbidly obese was associated with an EMR = 6.27 (95%CI 4.03–9.76) and a true pandemic interaction of almost 3 (2.87); the EMR declined at older ages.

S4 Table stratifies the model estimates for selected co-morbidities by age. Generally, most co-morbidities produced a higher EMR at younger ages, and a greater estimated pandemic interaction. For example, among 30–64-year-olds, Diabetes had an EMR = 6.36 (95%CI 4.72–8.56) and a true pandemic interaction of 3.07 (95%CI 2.22–4.23). For dementia, the EMR was 22.40 (95%CI 18.07–27.77) among the 65–79-year-olds. Mental health was the exception to this trend, where high EMR's persisted in the 80+ year olds (EMR = 3.88, 95%CI 2.86–5.26) and the estimated true impact of the pandemic increased with age. S5 Table stratifies the

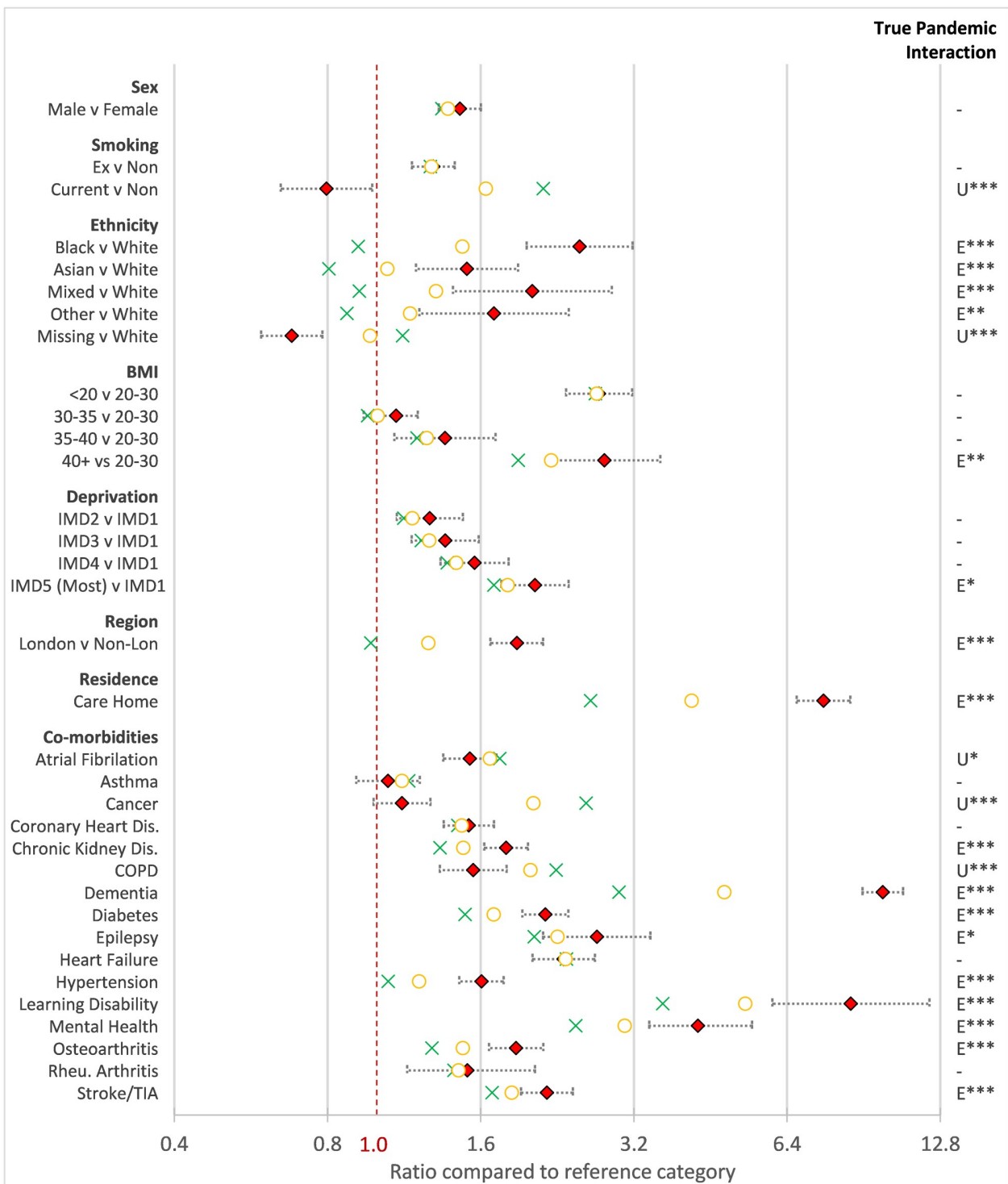

**Fig 1. Mortality ratios for 2020 (amber) and usual (green) with corresponding excess mortality ratio and 95%CI (red).** Green cross = Usual mortality ratio (2015–19), Amber circle = 2020 mortality ratio, Red Diamond = Excess mortality ratio. True Pandemic Interaction summaries statistical comparison between the Excess versus Usual mortality. E = Excess significantly higher, U = Usual significantly higher, * = p<0.05, ** = p<0.01, *** = p<0.001.

model estimates for dementia, learning disability and mental health by whether care home residence was recorded in the patient record. For all co-morbidities, the EMR and true pandemic interaction was higher in patients not recorded as living in a care home.

## Discussion

In the study, we have utilised a large electronic patient database to specifically study trends in excess mortality during the first wave of the COVID-19 pandemic in England. It confirmed many reported findings for risk factors such as age, obesity and ethnicity that were associated with dying from COVID-19 during this time, but also highlighted important differences, such as for current smokers and cancer patients, that would not be apparent from studying cause specific mortality on its own during the early stage of the pandemic.

### Rationale for excess mortality

Counting the number of deaths during a pandemic and comparing historically with deaths in similar non-pandemic periods is a robust methodology that has been used previously to provide international comparison of influenza deaths [18]. For COVID-19, first identified by Chinese authorities in January 2020, initial international comparisons have favoured this approach [1, 19] over simply counting COVID-19 deaths, due to the heterogeneity in how each country classified deaths from COVID-19. In England, this approach has been used to study the impact of community factors on excess mortality in an ecological analysis [20]. By analysing individual risk factors for excess mortality, and also by presenting the excess mortality ratios in the context of the usual mortality ratios, we think the methodology we have developed in studying the impact of these individual risk factors on excess mortality and identifying the true pandemic effect (interaction) for each risk factor provides a template that can be generalised across the COVID-19 pandemic, as well as to other causes of excess mortality such as influenza, and will be less susceptible to ecologic bias.

In England, virological testing was still building to capacity early in the pandemic [8], so early national comparisons made during these time are potentially biased by the lack of comprehensive COVID-19 testing at that time [21]. During the first wave it was estimated that over a fifth of excess deaths did not have a diagnosis of COVID-19 on their death certificate [2]. Thus, findings based on a diagnosis of COVID-19, may partially reflect risk factors for hospital admission, or for being tested for COVID-19, rather than risk factors for dying from COVID-19. Another advantage of studying excess mortality is that includes both in-patient and out-of-hospital deaths [22]. This is particularly relevant in the context of care-home COVID-related mortality during the first wave in England, where older patients were being routinely being discharged from hospitals to care homes without being testing for COVID-19 [21].

While emergency admissions have generally been increasing year-on-year in England they fell dramatically in April 2020 [23]. Thus, there were real concerns that one of the consequences of the "lockdown" measures used to contain COVID-19 in the population, could be an eventual increase in deaths from other causes [3], due to delays in presentation of non-COVID-19 medical emergencies such as heart attacks and strokes [4, 5]. Such excess deaths attributable to the pandemic, but not to COVID-19 itself, are captured as excess deaths in our analysis but would be missed by only counting COVID-19 deaths.

Another strength of our approach was to adopt an identical retrospective sampling approach by using the same set of practices in each period. This effectively helps to "cancel out" between practice differences in the recording of risk factors and other clinical information. While data linkage to national mortality data is available for CPRD [13], we chose to rely

on the recording based solely on the primary care record. Although the pattern of weekly excess mortality we estimated was similar to national figures for England (S1 Fig), there may be some data mis-classification with regards date of death by relying solely on the de-registration flag [17].

## Comparison with cause-specific findings

The most relevant comparison to our work is with the findings from the OpenSAFELY initiative [6, 7, 24], which built a large research dataset of general practices using the TPP SystmOne electronic health record system. Their initial report studied risk factors for death with COVID-19 as a certified cause up to early May 2020 (n = 5,683) among 17.4 million patients, and identified a series of risk factors for in-hospital death from COVID-19 [6], in particular people from Asian and black groups. A subsequent analysis using extended follow-up to the end of 2020 explored the ethnicity associations in more detail, finding the higher risks of both testing positive for SARS-CoV-2 and of experiencing an adverse COVID-19 outcome for non-white ethnicities, was not explained by sociodemographic and household characteristics [24]. Lastly, another OpenSAFELY analysis considered how the initial risk factors for COVID-19 death they identified, compared with those for non-COVID-19 mortality [7].

Our findings for ethnicity are broadly comparable with OpenSAFELY once the lower overall mortality risk in non-pandemic years is accounted for. Thus, when the authors compare their initial HR's for COVID-19 mortality for non-white ethnicities (1.4–1.5) to those found for non-COVID-19 mortality (<1), the estimated odds ratios for COVID-19 versus non-COVID-19 death among these ethnicities now exceed 2 [7]. Additionally, we also demonstrated how this impact among non-white ethnicities was more pronounced among people under 65 years, which may reflect greater employment in lower paid essential jobs which continued through the pandemic [25]. Evidence from the REACT-2 study suggested that the higher COVID-19 hospitalisation and mortality rates seen in minority ethnic groups may be a result of greater rates of infections, which were highest for the Black and Asian groups in the study less than 65 years old [26].

Our higher overall estimates for black ethnicities may be due to the different geographical coverages of the underlying GP software systems. The EMIS system (which CPRD Aurum is extracted from) has historically had greater reach in London which has a greater ethnic mix than other parts of England; London was where the greatest number of excess deaths were recorded during the first wave [2, 22]. The increased risk of death from COVID-19 in black, Asian and minority Ethnic groups in England was quickly identified during the early phase of the pandemic [25, 27], but there has been limited discussion and analysis on ethnic life expectancies prior to the pandemic [28] where the lower mortality rates in South Asian, Black and other minority groups have been attributed to a healthy migrant effect [29]. While approximately 20% of the patients in the study had no ethnicity recorded, this group had no excess mortality (EMR <1), suggesting our estimates for non-white ethnicities were unlikely to be exaggerated.

In England during the first wave, a consistent trend was observed in national data between mortality from COVID-19 and the Index of Multiple Deprivation [22]. The IMD is a composite measure of income, employment, education, health, crime, housing and the living environment used to summarise an individual's socio-economic position [14]. In our analysis, we estimated a significant trend with excess mortality for increasing levels of deprivation, with an approximately doubling in the excess morality risk in the most deprived quintile versus the least, which compared closely with the OpenSAFELY estimates for COVID-19 death, either age-sex or fully adjusted [6]. However, once the usual risks are accounted for, the risks are

attenuated, and a significant additional impact of the pandemic was only observed in the most deprived quintile (the true pandemic interaction we estimated was 1.21 compared to OpenSA-FELY OR = 1.29 vs. non-COVID-19 mortality [7]). Unlike OpenSAFELY [24], we were not able to explore the impact of household size since no household identifier was available in CPRD at the time we extracted our dataset.

Obesity has been shown to be associated with severe COVID-19 outcomes internationally [30], and while we observed higher excess mortality ratios for BMI>30, it was only among the morbidly obese (EMR = 2.80) where we estimated it was significantly higher than what would be usually observed. Among the specific co-morbidities we studied, the largest associations we saw with excess mortality were among patients with dementia (EMR = 9.9). This finding for dementia is likely intertwined with the failure to protect care home residents during the first wave of the pandemic in England [31], and we estimated a similar large excess mortality ratio (EMR = 7.5) for care home residence among patients with this recorded. If care home residents were dying from COVID-19 early on during the pandemic, but not being tested and recorded as COVID-19 deaths, this would explain why our estimate for excess mortality associated with dementia was much higher that the corresponding OpenSAFELY estimate (4.8) for Wave 1 for COVID-19 death [7]. A national study of provider-level administrative data on all care homes in England estimated that only 65% of excess deaths up to August 2020 were directly attributable to COVID-19 [32]. However, we need to be cautious around our findings regarding care homes as primary care recording via Read codes is incomplete; 1.4% of our patients aged 65+ years were estimated to be living in care homes, lower than recent reports (1.96–3.13%) from a similar database using more extensive methods [33] and from earlier census estimates. Thus, we cannot be certain that the inflated excess mortality risk among dementia patients *without* any recording of care home residence are all from community-based patients.

Patients with a learning disability are already at a known higher risk of respiratory associated death than in the general population [34], and we estimated an EMR = 8.5, which was more than a doubling of the usual mortality risk. This compares with another OpenSAFELY analysis, that estimated of HR = 8.2 for COVID-19 death [35]. The finding of a four times higher risk of excess mortality among patients with severe mental illness (psychosis, schizophrenia and bipolar affective disorder) has not, to our knowledge, previously been shown. However, there have been considerable reductions in primary care-recorded mental illness and related consultations during 2020 [4, 36], and survey data has shown that adults with pre-existing mental health problems had worse mental health outcomes during the pandemic [37]. For diabetes, we estimated an approximate doubling of the risk for excess mortality (EMR = 2.1), which compares with an OR = 2.03 for in-hospital death from COVID-19 during May to March 2020 among type 2 diabetes in a complete population analysis (61 million) [38]. The same study found greater risk among type 1 diabetics, which may reflect the greater associations we found among the younger (30–64 years) people with diabetes in our analysis. Another analysis of CPRD data observed a dramatic fall in contacts for diabetic emergencies after the introduction of population restrictions in March 2020 [4].

Not all risk factors were positively associated with the pandemic. We found that current smokers, and patients with a history of cancer or COPD all had estimated mortality ratios in 2020 which were significantly lower than their usual mortality. The absence of an association with COVID-19 mortality among current smokers and some types of cancer (non-haematological) was first observed in OpenSAFELY [6], but when the risk of COVID-19 versus non-COVID-19 death was compared, odds ratios <1 were estimated [7], which parallels our finding of a potentially reduced impact of the pandemic within these groups. For cancer, the results appear counter-intuitive as it is generally assumed cancer survivors are a high-risk group for

severe COVID-19 outcomes [39]. This may reflect the raised risk for COVID-19 mortality among haematological malignancies [6], which directly impact the immune system [40]. While we did observe a higher excess mortality among haematological cancers (S6 Table), the reduced impact of the pandemic was observed for both groups of cancers (true pandemic interactions <1), which matches what was found in the OpenSAFELY direct comparison of COVID-19 versus non-COVID-19 death where, although the risk was much less among non-haematological cancers, the OR's were still below 1 for haematological cancers [7]. These findings suggest that across the first wave in England, cancer survivors were able to lessen the full effect of the pandemic, perhaps through shielding / reduced social interaction and less contact with healthcare over this period [40].

The suggestion that perceived high-risk groups were able to mitigate risk could also explain the findings we observed among COPD patients, where one might have expected higher excess mortality from the excess lung damage caused by COVID-19 [41]. However early studies from China reported lower than expected prevalence of asthma and COPD in patients diagnosed with COVID-19 [42], which prompted some speculation that inhaled corticosteroids may have a protective role to play. While the evidence of a beneficial effect among COPD patients using observational data in the UK has been mixed [43, 44], recent trials of inhaled budesonide among all people with suspected or mild COVID-19 in the community have shown improvements in time to recovery [45, 46] and potentially lower rates of hospital admissions or death [46].

The observation that there was no excess mortality among current smokers in our study, tallies with the OpenSAFELY finding of no association between current smoking and COVID-19 mortality [6]. If true, it does suggest that among the excess mortality during the first wave in the UK, the component attributable to smoking-related outcomes such as cardiovascular fatalities were negligible. However, there may still be long-term complications for patients with coronary heart disease resulting from the reduction in expected hospitalisations during the first wave [5]. Elsewhere, early studies mainly from China observed lower than expected prevalence of current smoking among patients hospitalised with COVID-19 [47], and a living review has concluded that "compared with never smokers, current smokers appear to be at reduced risk of SARS-CoV-2 infection" [48]. However in the UK, the Zoe COVID-19 symptom study showed that during March-April 2020 current smokers were at an increased risk of developing symptomatic COVID-19 [49]. Another explanation could be that smokers are protected from the most severe impact of COVID-19 [50], but the UK Biobank study has suggested that once infected, older smokers were twice as likely to die from COVID-19 than never smokers [51].

## Conclusion

In conclusion, we have demonstrated how focussing on excess mortality during the early stages of the COVID-19 pandemic in England can provide novel insights and robust estimates of mortality risk that account for the usual trends in the population. This approach removes the need for complex adjustment of confounders and allows the impact of a pandemic to be studied without specifically identifying deaths due to a specific cause.

## Supporting information

**S1 Appendix. Defining usual mortality ratio, excess mortality ratio and true pandemic interaction.**
(PDF)

**S1 Fig. Weekly excess mortality during 2020 using CPRD versus ONS national figures for England.**
(PDF)

**S1 Checklist. STROBE statement—Checklist of items that should be included in reports of observational studies.**
(DOC)

**S1 Table. Number of total patients and recorded deaths in 2020 and 2015–9.**
(PDF)

**S2 Table. Mortality ratios for 2020 and 2015–9 (Usual) with corresponding excess mortality ratio (EMR) and true pandemic interaction (TPI) from mutually adjusted models†.**
(PDF)

**S3 Table. Mortality ratios for 2020 and 2015–9 (Usual) with corresponding excess mortality ratio (EMR) and true pandemic interaction (TPI) for sex, smoking, ethnicity, deprivation, BMI and region stratified by age.**
(PDF)

**S4 Table. Mortality ratios for 2020 and 2015–9 (Usual) with corresponding excess mortality ratio (EMR) and true pandemic interaction (TPI) for selected co-morbidities stratified by age.**
(PDF)

**S5 Table. Mortality ratios for 2020 and 2015–9 (Usual) with corresponding excess mortality ratio (EMR) and true pandemic interaction (TPI) for selected co-morbidities stratified by care home residence.**
(PDF)

**S6 Table. Mortality ratios for 2020 and 2015–9 (Usual) with corresponding excess mortality ratio (EMR) and true pandemic interaction (TPI) for haematological and non- haematological cancer.**
(PDF)

## Acknowledgments

IC, DC and DS conceptualised the study. IC was curated the datasets and carried out the formal analysis. All authors (IC, DC, TH, SW, UC, DS) contributed writing and editing of the submitted manuscript.

This study is based in part on data from the Clinical Practice Research Datalink obtained under licence from the UK Medicines and Healthcare products Regulatory Agency. The data is provided by patients and collected by the NHS as part of their care and support. The interpretation and conclusions contained in this study are those of the author/s alone.

## Author Contributions

**Conceptualization:** Iain M. Carey, Derek G. Cook, David P. Strachan.

**Data curation:** Iain M. Carey.

**Formal analysis:** Iain M. Carey.

**Methodology:** Iain M. Carey, Derek G. Cook, David P. Strachan.

**Writing – original draft:** Iain M. Carey, Derek G. Cook, Tess Harris, Stephen DeWilde, Umar A. R. Chaudhry, David P. Strachan.

**Writing – review & editing:** Iain M. Carey, Derek G. Cook, Tess Harris, Stephen DeWilde, Umar A. R. Chaudhry, David P. Strachan.

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
