## [Decision Letter · Decision Letter 0]

28 Sep 2021

PONE-D-21-27684Comparison of risk factors for excess all-cause mortality during the first wave of the COVID-19 pandemic in England (March-May 2020) compared to usual all-cause mortality during 2015-2019: a retrospective cohort study of primary care dataPLOS ONE

Dear Dr. Carey,

Thank you for submitting your manuscript to PLOS ONE. After careful consideration, we feel that it has merit but does not fully meet PLOS ONE’s publication criteria as it currently stands. Therefore, we invite you to submit a revised version of the manuscript that addresses the points raised during the review process.

We look forward to receiving your revised manuscript.

Kind regards,

Shinya Tsuzuki, MD, MSc

Academic Editor

PLOS ONE

Journal Requirements:

Additional Editor Comments:

Several reviewers made a favorable assessment on the manuscript then only a few minor comments should be addressed before publication.

Reviewers' comments:

Reviewer's Responses to Questions

**Comments to the Author**

1. Is the manuscript technically sound, and do the data support the conclusions?

Reviewer #1: Yes

Reviewer #2: Yes

Reviewer #3: Yes

Reviewer #4: Yes

2. Has the statistical analysis been performed appropriately and rigorously? 

Reviewer #1: Yes

Reviewer #2: Yes

Reviewer #3: Yes

Reviewer #4: Yes

3. Have the authors made all data underlying the findings in their manuscript fully available?

Reviewer #1: Yes

Reviewer #2: Yes

Reviewer #3: Yes

Reviewer #4: Yes

4. Is the manuscript presented in an intelligible fashion and written in standard English?

Reviewer #1: Yes

Reviewer #2: Yes

Reviewer #3: Yes

Reviewer #4: Yes

5. Review Comments to the Author

Reviewer #1: The excess mortality in England during the first wave of COVID-19 pandemic, compared with the mortality during 2015-19 was studied. Mortality rates in among different population groups for the period from 18th March to 19th May 2020 was compared with similar indicators in 2015 – 2019.

It has been shown that the main risk factors were age >80, non-white ethnicity, dementia, learning disability. By contrast, smokers, and patients with a history of cancer or Chronic Obstructive Pulmonary Disease had mortality ratios in 2020 significantly lower than their usual mortality.

Reviewer #2: I think that such articles are important and valuable in order to understand the extent of mortality caused by the Covid-19 epidemic. In this respect, this article presents well-designed and versatile data.

Reviewer #3: This article provides important insight into changes in all-cause mortality during, and possibly due to, the COVID-19 pandemic in England in early 2020.

Specific feedback below:

Title:

The title of the paper is probably longer and more detailed than necessary and could be contracted. For example, “Risk factors for excess all-cause mortality during the first wave of the

COVID-19 pandemic in England” might adequately convey the content in a more succinct manner

Abstract:

The somewhat strange use of active voice in the abstract is off-putting e.g. “Patients had their [...] status extracted” and “Poisson regression modelled” under the methods heading. The author’s intermittently use active “we” and passive voice throughout the rest of the manuscript in the manuscript. Maintaining consistency might improve the overall flow.

Methods:

Line 88: The acronym “EMIS” needs to be defined

Line 98: the authors state they selected a subset of 770 practices providing data to CPRD.

It would be useful to note what proportion of all practices participating in CPRD this represents, and how representative these practices are of all practices in CPRD (Noting representativeness of CPRD compared to all general practices is discussed later).

Line 105: The authors should provides justification for selecting the age range 30-104 years i.e. why were younger adults or children not included?

Results:

The meaning of the symbols in Figure 1 probably don’t need to be defined in the text as they are given in the figure legend.

Discussion:

Given almost 20% of patients did not have ethnicity recorded, it would worth commenting on effect of this missing data on validity of ethnicity estimates.

Line 288: Sentence beginning:“For COVID-19, a novel coronavirus” The meaning of this sentence is not clear to me, also COVID-19 is the disease, SARS-CoV-2 is the novel coronavirus.

Reviewer #4: This is a very interesting manuscript where the authors use a large electronic primary care database to estimate the impact of risk factors on excess mortality in England during the first wave, compared with the mortality during 2015-19. The manuscript is very well written and utilizes sound statistical methods which are very well described. Here are my two comments for the authors to consider:

1. In the abstract, the authors state that “Our approach illustrates the use of a novel methodology for evaluating a 52 pandemic’s impact without requiring cause-specific mortality data.”. It is not entirely clear to me what is novel in this analysis compared to other methods used in the past. Can the authors be explicit in stating what is novel in their methods? As far as I know, similar analytic approaches have been used for years to estimate excess mortality associated with influenza infection.

2. Could the authors provide a comment/justification for why the considered 5 pre-pandemic years (2015-2019 combined) rather than less, say 2 or 3 most recent year? Was it all necessary if they could have analyzed the data with less?

6. PLOS authors have the option to publish the peer review history of their article (what does this mean?). If published, this will include your full peer review and any attached files.

Reviewer #1: No

Reviewer #2: **Yes: **Ali Acar

Reviewer #3: **Yes: **Christopher Bailie

Reviewer #4: No

---

## [Author Response · Author response to Decision Letter 0]

22 Oct 2021

Additional Editor Comments:

Several reviewers made a favorable assessment on the manuscript then only a few minor comments should be addressed before publication.

* We thank all four reviewers for their positive feedback and address the minor comments they raised below. Where the text has changed, we have indicated below.

Reviewers' comments:

Reviewer's Responses to Questions

Comments to the Author

* Sections 1 to 4 – No comments necessary

5. Review Comments to the Author

Reviewer #1: The excess mortality in England during the first wave of COVID-19 pandemic, compared with the mortality during 2015-19 was studied. Mortality rates in among different population groups for the period from 18th March to 19th May 2020 was compared with similar indicators in 2015 – 2019.

It has been shown that the main risk factors were age >80, non-white ethnicity, dementia, learning disability. By contrast, smokers, and patients with a history of cancer or Chronic Obstructive Pulmonary Disease had mortality ratios in 2020 significantly lower than their usual mortality.

Reviewer #2: I think that such articles are important and valuable in order to understand the extent of mortality caused by the Covid-19 epidemic. In this respect, this article presents well-designed and versatile data.

Reviewer #3: This article provides important insight into changes in all-cause mortality during, and possibly due to, the COVID-19 pandemic in England in early 2020.

Specific feedback below:

Title:

The title of the paper is probably longer and more detailed than necessary and could be contracted. For example, “Risk factors for excess all-cause mortality during the first wave of the

COVID-19 pandemic in England” might adequately convey the content in a more succinct manner

* We agree the title could be improved by shortening. However, to comply with STROBE we have retained the last section to give a new title of 

“Risk factors for excess all-cause mortality during the first wave of the COVID-19 pandemic in England: a retrospective cohort study of primary care data”

Abstract:

The somewhat strange use of active voice in the abstract is off-putting e.g. “Patients had their [...] status extracted” and “Poisson regression modelled” under the methods heading. The author’s intermittently use active “we” and passive voice throughout the rest of the manuscript in the manuscript. Maintaining consistency might improve the overall flow.

*We apologise for this oversight. In the abstract we now say

“Medical history, ethnicity, area-based deprivation and vital status data were extracted for an average of 4.8 million patients aged 30-104 years for each year between 18-March and 19-May over a 6-year period (2015-2020). We used Poisson regression to model total mortality adjusting for age and sex, with interactions between each RF and period (pandemic vs. 2015-19). Total pandemic mortality was partitioned into "usual" and "excess" components, assuming 2015-19 rates represented "usual" mortality. The association of each RF with the 2020 "excess" component was derived as the excess mortality ratio (EMR) and compared with the usual mortality ratio (UMR).”

Methods:

Line 88: The acronym “EMIS” needs to be defined

*We now say

Egton Medical Information Systems

Line 98: the authors state they selected a subset of 770 practices providing data to CPRD.

It would be useful to note what proportion of all practices participating in CPRD this represents, and how representative these practices are of all practices in CPRD (Noting representativeness of CPRD compared to all general practices is discussed later).

*The August 2020 extract of CPRD Aurum consisted of 1366 practices in total, so the 770 we chose represents 56%. CPRD updates the extracts on a monthly basis and not all practices provide data to the most recent date. As we chose 1/8/2020 as a cut-off date for practices to be included to allow enough time for death de-registrations to be recorded, a large proportion of practices were excluded as they only had data to June or July 2020 at this time. There doesn’t appear to be any systematic reason why these practices would differ, and the geographical distribution is similar. We do not have details on the patients from the non-770 practices. 

We think the issue of representativeness is how geographically representative our 770 practices are of all practices rather than the CPRD itself. We already note in the discussion that we have an oversampling of London practices due to the earlier pattern of excess mortality. However, the key aspect is that our analysis uses the same 770 practices for all years to compare mortality within practices. 

We have added the following sentence to the methods

"A total of 770 (56%) practices were identified, with exclusions due to data not available to August 2020 or no linkage available".

Line 105: The authors should provides justification for selecting the age range 30-104 years i.e. why were younger adults or children not included?

*We now say

“We further restricted to adults aged between 30 and 104 years old, as there would be little excess mortality in the young as well as incomplete data for many risk factors, and also excluded a small number of patients (<1%) without linkage to IMD.”

Results:

The meaning of the symbols in Figure 1 probably don’t need to be defined in the text as they are given in the figure legend.

*We have now removed these

Discussion:

Given almost 20% of patients did not have ethnicity recorded, it would worth commenting on effect of this missing data on validity of ethnicity estimates.

In UK primary care data, ethnicity is more complete in patients more recently registered (Mathur et al, Journal of Public Health 2014 36(4): 684-692), which suggests that the missing group may be overrepresented by long-term registered patients. However, the group will also likely contain a significant portion of healthier patients, who are not being regularly seen at their practice and having their ethnicity recorded. 

*For each annual cohort, we only used information recorded up to that point in time. This means that the missing % for ethnicity was largely stable in each annual cohort, and as a result one could assume that the (unknown) ethnicity mix among the missing was broadly similar between 2015-9 and 2020. Therefore, our estimates represent the risk of mortality for patients being recorded with ethnicity X in 2020 vs. the risk for patients being recorded with the same ethnicity X in 2015-9. When this classification was “missing”, the excess mortality associated with this was less than 1 (Figure 1, Table 2) compared to the white baseline group. Thus, the estimates we produced for recorded non-white groups (vs. recorded white) may be conservative if anything, and are unlikely to have been exaggerated. 

We have added this sentence to the discussion

“While approximately 20% of the patients in the study had no ethnicity recorded, this group had no excess mortality (EMR <1), suggesting our estimates for non-white ethnicities were unlikely to be exaggerated.”

Line 288: Sentence beginning: “For COVID-19, a novel coronavirus” The meaning of this sentence is not clear to me, also COVID-19 is the disease, SARS-CoV-2 is the novel coronavirus.

*We have simplified sentence to

 “For COVID-19, first identified by Chinese authorities in January 2020, initial international comparisons have favoured this approach[1,19] over simply counting COVID-19 deaths, due to the heterogeneity in how each country classified deaths from COVID-19”.

Reviewer #4: This is a very interesting manuscript where the authors use a large electronic primary care database to estimate the impact of risk factors on excess mortality in England during the first wave, compared with the mortality during 2015-19. The manuscript is very well written and utilizes sound statistical methods which are very well described. Here are my two comments for the authors to consider:

1. In the abstract, the authors state that “Our approach illustrates the use of a novel methodology for evaluating a 52 pandemic’s impact without requiring cause-specific mortality data.”. It is not entirely clear to me what is novel in this analysis compared to other methods used in the past. Can the authors be explicit in stating what is novel in their methods? As far as I know, similar analytic approaches have been used for years to estimate excess mortality associated with influenza infection.

We agree the brief sentence in the abstract failed to make an adequate case; that our study is of individual risk factors in relation to excess mortality which is novel. 

*We have changed the abstract sentence to

“Our approach illustrates a novel methodology for evaluating a pandemic’s impact by individual risk factor, without requiring cause-specific mortality data”

We have also added an additional sentence in the discussion, with a new reference (Davies et al., Nature Communications 2021 12(1): 3755) which also studied excess mortality, but only looked at community, not individual factors, using an ecological approach.

“In England, this approach has been used to study the impact of community factors on excess mortality[20] in an ecological analysis. By analysing individual risk factors for excess mortality, and also by presenting the excess mortality ratios in the context of the usual mortality ratios, we think the methodology we have developed for studying the impact of individual risk factors on excess mortality and identifying the true pandemic effect (interaction) for each risk factor. It provides a template that can be generalised across the COVID-19 pandemic, as well as to other causes of excess mortality such as influenza, and will be less susceptible to ecologic bias.”

2. Could the authors provide a comment/justification for why the considered 5 pre-pandemic years (2015-2019 combined) rather than less, say 2 or 3 most recent year? Was it all necessary if they could have analyzed the data with less?

*Using 5 years for the reference period is commonly used when assessing excess mortality (e.g., Davies et al., Nature Communications 2021 12(1): 3755) and would produce more precise estimates than using 2 or 3 years.

---

## [Decision Letter · Decision Letter 1]

3 Nov 2021

PONE-D-21-27684R1Risk factors for excess all-cause mortality during the first wave of the COVID-19 pandemic in England : a retrospective cohort study of primary care dataPLOS ONE

Dear Dr. Carey,

Thank you for submitting your manuscript to PLOS ONE. After careful consideration, we feel that it has merit but does not fully meet PLOS ONE’s publication criteria as it currently stands. Therefore, we invite you to submit a revised version of the manuscript that addresses the points raised during the review process.

We look forward to receiving your revised manuscript.

Kind regards,

Shinya Tsuzuki, MD, MSc

Academic Editor

PLOS ONE

Journal Requirements:

Additional Editor Comments:

Please modify the manuscript according to the reviewer's comments before publipcation.

Reviewers' comments:

Reviewer's Responses to Questions

**Comments to the Author**

1. If the authors have adequately addressed your comments raised in a previous round of review and you feel that this manuscript is now acceptable for publication, you may indicate that here to bypass the “Comments to the Author” section, enter your conflict of interest statement in the “Confidential to Editor” section, and submit your "Accept" recommendation.

Reviewer #3: All comments have been addressed

2. Is the manuscript technically sound, and do the data support the conclusions?

Reviewer #3: Yes

3. Has the statistical analysis been performed appropriately and rigorously? 

Reviewer #3: Yes

4. Have the authors made all data underlying the findings in their manuscript fully available?

Reviewer #3: Yes

5. Is the manuscript presented in an intelligible fashion and written in standard English?

Reviewer #3: Yes

6. Review Comments to the Author

Reviewer #3: Previous comments have been adequately addressed.

The Authors may wish to re-revise changes to the two sentences from lines 291-296 beginning "By analysing individual risk factors...", as their meaning is currently unclear.

7. PLOS authors have the option to publish the peer review history of their article (what does this mean?). If published, this will include your full peer review and any attached files.

Reviewer #3: No

---

## [Author Response · Author response to Decision Letter 1]

4 Nov 2021

We apologise for the minor punctuation error where a full stop had not been marked for deletion.

The corrected sentence now reads

“By analysing individual risk factors for excess mortality, and also by presenting the excess mortality ratios in the context of the usual mortality ratios, we think the methodology we have developed for studying the impact of individual risk factors on excess mortality and identifying the true pandemic effect (interaction) for each risk factor provides a template that can be generalised across the COVID-19 pandemic, as well as to other causes of excess mortality such as influenza, and will be less susceptible to ecologic bias”

---

## [Editor Report · Decision Letter 2]

9 Nov 2021

Risk factors for excess all-cause mortality during the first wave of the COVID-19 pandemic in England : a retrospective cohort study of primary care data

PONE-D-21-27684R2

Dear Dr. Carey,

We’re pleased to inform you that your manuscript has been judged scientifically suitable for publication and will be formally accepted for publication once it meets all outstanding technical requirements.

Kind regards,

Shinya Tsuzuki, MD, MSc

Academic Editor

PLOS ONE
---

## [Editor Report · Acceptance letter]

1 Dec 2021

PONE-D-21-27684R2 

Risk factors for excess all-cause mortality during the first wave of the COVID-19 pandemic in England : a retrospective cohort study of primary care data 

Dear Dr. Carey:

I'm pleased to inform you that your manuscript has been deemed suitable for publication in PLOS ONE. Congratulations! Your manuscript is now with our production department. 

Kind regards, 

on behalf of

Dr. Shinya Tsuzuki 

Academic Editor

PLOS ONE